# β-Synuclein: An Enigmatic Protein with Diverse Functionality

**DOI:** 10.3390/biom12010142

**Published:** 2022-01-16

**Authors:** Junna Hayashi, John A. Carver

**Affiliations:** Research School of Chemistry, The Australian National University, Acton, ACT 2601, Australia; junna.hayashi@anu.edu.au

**Keywords:** synuclein, molecular chaperone, neurodegeneration, Parkinson’s disease, dementia with Lewy bodies

## Abstract

α-Synuclein (αS) is a small, unstructured, presynaptic protein expressed in the brain. Its aggregated form is a major component of Lewy bodies, the large proteinaceous deposits in Parkinson’s disease. The closely related protein, β-Synuclein (βS), is co-expressed with αS. In vitro, βS acts as a molecular chaperone to inhibit αS aggregation. As a result of this assignation, βS has been largely understudied in comparison to αS. However, recent reports suggest that βS promotes neurotoxicity, implying that βS is involved in other cellular pathways with functions independent of αS. Here, we review the current literature pertaining to human βS in order to understand better the role of βS in homeostasis and pathology. Firstly, the structure of βS is discussed. Secondly, the ability of βS to (i) act as a molecular chaperone; (ii) regulate synaptic function, lipid binding, and the nigrostriatal dopaminergic system; (iii) mediate apoptosis; (iv) participate in protein degradation pathways; (v) modulate intracellular metal levels; and (vi) promote cellular toxicity and protein aggregation is explored. Thirdly, the P123H and V70M mutations of βS, which are associated with dementia with Lewy bodies, are discussed. Finally, the importance of post-translational modifications on the structure and function of βS is reviewed. Overall, it is concluded that βS has both synergistic and antagonistic interactions with αS, but it may also possess important cellular functions independent of αS.

## 1. Intrinsically Disordered Proteins

Intrinsically disordered proteins (IDPs) are characterized by their inability to fold into a stable or well-defined three-dimensional structure [1]. IDPs and intrinsically disordered regions in proteins account for more than one-third of the human proteome [2]. Their abundance implies their importance in key cellular processes such as homeostasis and survival [3,4]. The absence of a defined structure confers IDPs with conformational flexibility, allowing them to partake in dynamic and transient, often multivalent, molecular interactions. IDPs are often regulatory proteins, whereby they function as protein interaction network hubs. For example, they control signaling pathways, the regulation of transcription and translation, and the cell cycle, all of which require rapid interactions with high specificity and modest affinity to numerous targets according to the cell’s dynamic requirements. Post-translational modifications (PTMs) are frequently employed to offer an additional layer of fine-tuning in these processes. Disordered regions often contain multiple conserved and repeat sequence motifs such as amphipathic and short linear motifs which mediate binding and low-sequence complexity or prion-like sequences which regulate cellular compartmentalization. IDPs usually lack amino acid residues which promote order, such as cysteine and asparagine, as well as those which stabilize and form the hydrophobic core of folded, globular proteins including hydrophobic aromatic residues. On the other hand, they frequently contain a high content of polar and structure-breaking residues (e.g., proline) that facilitate disorder [5].

Considering their abundance and their involvement in crucial biological processes, mutations in these proteins or regions are linked to disease, with approximately 20% of human disease mutations occurring within intrinsically disordered regions of proteins [6].

## 2. The Synuclein Family: α-, β-, and γ-Synuclein

The synucleins are IDPs. They were first discovered in the electric organ *Torpedo*, in which co-sedimentation analysis revealed their localization in the synaptic vesicles of neurons. They were given the name ‘synucleins’ due to their association with the synaptic vesicles and the nuclear membrane, although the latter is not frequently observed [7]. In humans, α-, β-, and γ-synuclein (herein referred to as αS, βS, and γS) are expressed in the early development of neurons in the substantia nigra, pointing to a functional role in neuronal development and maturation [8]. Moreover, the synuclein proteins are absent in invertebrates, suggesting that they are not implicated in basic cellular processes, but have a higher order function such as synaptic plasticity. In parallel, mice which lacked one or two synucleins are fertile and exhibit similar life spans to their wild-type (WT) counterparts [9,10]. Varied findings have been reported for mice which lack all three synucleins, which suggest that breeding patterns and other environmental factors play a role in their development [9,11]. Of the three synucleins, αS and βS have greater sequence similarity between themselves than with γS (Figure 1). The primary amino acid sequence of synucleins exhibits a tripartite organization, with a highly conserved N-terminal region, the central hydrophobic non-amyloid β component (NAC), and a highly acidic C-terminal region. βS lacks most of the NAC region. 

βS accounts for 75–80% of the total synuclein mRNA pool in the human body, whereas αS is the most abundant synuclein protein. αS is expressed predominantly in the neurons of the central nervous system and in red blood cells. βS is expressed in the neurons of the central nervous system, olfactory receptor neurons in the olfactory epithelium [14], skeletal muscle [15], Sertoli cells, astrocytes [16], and myelin [17]. γS is expressed in adipose and peripheral neuronal tissues [18]. γS regulates kinase activity, cell cycle checkpoints, and has been found to be overexpressed in cancer [19]. Although each synuclein possesses independent roles within the cell, they can, at least partially, compensate for compromised or lost function amongst themselves. Throughout most of the body, αS and βS are co-expressed, leading to the hypothesis that these proteins could be cooperative or antagonistic in their cellular functions.

In 1997, Spillantini et al. reported that αS, but not βS or γS, was the major component in Lewy bodies, the proteinaceous deposits within the brain that are a pathological hallmark of Parkinson’s disease (PD) [20]. Over the past 25-odd years, a myriad of investigations has been undertaken into the structure and function of the synucleins in an attempt to understand their role in PD and related diseases.

## 3. The Structure of β-Synuclein

βS is 134 amino acids long, with a high content of the non-polar aliphatic amino acid alanine (13.4%) and the negatively charged glutamic acid (18.7%). βS lacks cysteine and tryptophan residues (Figure 1). At neutral pH, secondary structure analysis using far UV-circular dichroism (UV-CD) spectroscopy revealed that βS adopts an unstructured conformation, showing a negative ellipticity around 196 nm (consistent with an absence of a defined secondary structure) and the absence of minima between 210 and 230 nm [21]. The Stokes radius, which reflects the hydrodynamic radius as determined by size exclusion chromatography, matched the theoretical value for an unfolded polypeptide [21]. In comparison to αS and γS, small angle neutron scattering, size exclusion chromatography [21], and pulse field gradient NMR spectroscopy [22] indicated that βS exhibits a more extended conformation. Near UV-CD revealed that no tertiary structure was present [23], consistent with the intrinsically disordered property of βS. Furthermore, Raman optical activity, a technique which measures the difference in intensity of Raman scattering from chiral molecules in left- and right-circularly polarized light, revealed that βS adopts a significant amount of a polyproline type-II (PP-II) helical conformation [24], as is common for IDPs. The PP-II helix in IDPs may act as a template for the formation of an amyloid fibrillar conformation [24]. The NAC region in αS is crucial to its amyloid fibril formation, as occurs for the protein within Lewy bodies. The absence of most of the NAC region in βS (Figure 1) is a major determinant in the protein’s inability to form amyloid fibrils under physiological conditions. Finally, βS exists primarily as a tetramer in the primary neurons of rats [25], although in vitro studies suggest that it is predominantly monomeric [23]. The intrinsically disordered nature of βS means that it can adopt multiple dynamic conformations, properties which underlie its various functions within the cell.

## 4. The Function of β-Synuclein

### 4.1. Molecular Chaperone Ability of β-Synuclein

The ability of βS to inhibit the amyloid fibrillar aggregation of αS has been well established [21,23,26,27,28,29,30,31]. In vitro, the molecular chaperone action of βS is achieved by elongating the lag phase of αS aggregation [23,26,28,32,33], inhibiting secondary nucleation, and diluting the amount of αS at the lipid surface during lipid-induced aggregation of αS [26]. In a cell-free αS and βS co-expression system, fluorescent time traces revealed that the titration of βS led to a reduction in the size of αS aggregates, with an equimolar concentration of βS resulting in only the monomeric form of αS being present [31]. On the contrary, another study using TEM revealed that an equimolar concentration of αS and βS led to shorter and more branched amyloid fibrils of αS [29]. Mass spectrometry of the dissolved fibrils formed in βS-treated αS aggregation revealed that βS was not incorporated into the fibrils [21]. Single molecule fluorescence experiments also revealed that βS can interact with monomeric and oligomeric αS, but not the fibrillar form [31]. Furthermore, aggregation inhibition curves generated by fluorescently tagged αS and βS demonstrated that βS can better prevent the aggregation of A30P and G51D αS, which are less aggregation prone in comparison to the rapidly aggregating and fibril forming αS mutants, E46K, H50Q, and A53T [31], suggesting that βS is less effective at suppressing the aggregation of rapidly fibril-forming αS species. Overall, these results imply that during chaperone action, βS predominantly acts at the initial stages of the αS aggregation process to prevent αS from amyloid fibril formation. The interaction is transient and dynamic. The characteristics of αS chaperone action are comparable to those of ‘holdase’ molecular chaperones, such as the intramolecular small heat-shock proteins (sHsps) [32,33].

To understand the underlying molecular mechanism of βS inhibition of αS aggregation, their interactions have been investigated. To determine the residue-specific interaction in vitro between αS and βS, NMR paramagnetic relaxation experiments concluded that monomeric βS inhibits αS aggregation via its C-terminal residues, E115-A134, binding to the N-terminal residues, L38-K45, of αS, leading to heterodimer formation [28]. Importantly, G36-S42 in αS is required for the protein to aggregate in vitro and in vivo [34]. The strength of the interaction between αS and βS was quantified with a K_D_ value of ~100 μM over a range of 40–350 μM αS in comparison to the equivalent N-terminal region of αS binding to the C-terminal residues, Y125-A140, of αS, with a K_D_ value of ~500 μM over a range of 40–350 μM, i.e., the αS/βS heterodimer binds with an approximately five times higher affinity than the αS/αS homodimer [28]. The αS homodimer may be more prone to aggregation, as weaker interactions would allow for conformational rearrangement to an in-register parallel cross-β-sheet structure that is a prerequisite for amyloid fibril formation. Furthermore, the higher affinity between βS and αS and the interaction over a wider range of βS C-terminal residues may prevent the conformational rearrangement and inhibit the association of αS with itself, hence preventing αS aggregation [28]. A separate study quantified the intermolecular electrostatic energy to form αS/αS homo- or αS/βS heterodimers [30]. The heterodimers had a substantially lower minimum energy of −31.6 kcal mol^−1^ in comparison to the homodimer of −13.4 kcal mol^−1^, implying that heterodimer formation is much more favorable [30]. Moreover, molecular docking stimulations also revealed that βS can interact with the αS dimer, preventing the binding of a subsequent αS monomer to the homodimer, thereby inhibiting propagation and aggregation of αS [30]. Single molecule fluorescence experiments in a cell-free system also suggested that a gradual increase in the concentration of βS replaces the small oligomers formed by αS, thereby preventing interactions between αS to inhibit self-assembly and commence amyloid fibril formation [31]. These studies are complementary in their conclusion that through the favorable interaction of βS with αS, βS prevents the association of αS molecules early along the αS aggregation cascade, thereby enabling βS to function as an effective molecular chaperone.

In addition, βS prevents αS-related toxicity in a cellular environment. For example, when the human neuroblastoma SH-SY5Y cell line was transfected with βS and treated with copper-induced neurotoxic αS oligomers, the cells were significantly resistant to the cytotoxic effects of the αS oligomers and had lower levels of reactive oxygen species in comparison to the non-transfected controls [35]. Moreover, when the human embryonic kidney HEK293T cells, which have neuronal-like properties [36], were transfected with αS or βS, cells expressing βS had decreased plasma membrane ion permeability in comparison to those expressing αS [37]. Furthermore, βS prevented αS-related toxicity upon modulation of the proteasome, a protein complex which degrades unwanted or damaged proteins in the cell. For example, PD brains also exhibited a decrease in proteasomal activity, suggesting a role of the proteasome in the pathogenesis of neurodegenerative diseases [38]. When aggregated αS was incubated with the cell lysate of HEK293 cells, the activity of the 26S ubiquitin-independent proteasome was inhibited. However, with the addition of recombinant βS prior to incubation with aggregated αS, proteasomal activity was largely intact [39]. Taken together, these findings are consistent with βS protecting cells against the neurotoxic effects induced by αS.

βS also prevents αS-related toxicity in whole organisms. Double-transfected αS and βS mice displayed a 40% decrease in αS immunoreactivity in neuronal inclusions and did not exhibit impairment in motor function compared to transgenic mice transfected with only αS [40,41]. αS-transgenic mice injected with βS lentivirus led to a reduced amount of αS in inclusion bodies [42]. Similarly, when A53T M83 mice, a PD mouse model which develops a severe and complex motor phenotype, were crossed with mice overexpressing βS, there was a significant reduction in αS aggregation with fewer motor deficits and an increase in life span in comparison to A53T M83 mice [37]. Moreover, a peptide encompassing G36-R45 of βS was engineered to evade degradation by proteases and increase its stability by retro-inversion and modifying its amino acids into their d-enantiomer. The peptide was then used to treat an A53T αS *Drosophila melanogaster* PD model. At day 27, non-treated flies were immobile in comparison to their peptide-treated counterpart in which 86% resembled the motor abilities of WT flies [43]. Peptide-treated flies also had a reduced amount of αS in their brains in comparison to their non-treated counterpart [43]. It is concluded that βS can prevent αS-induced neurotoxicity in mice and *Drosophila melanogaster*.

In vitro, βS acts as a molecular chaperone to prevent the aggregation of other proteins. At a 6 and 12 molar excess, βS suppressed the temperature-induced amorphous aggregation of aldolase by 46% and 80%, respectively [44]. A six molar excess of βS led to a 90% reduction in temperature-induced amorphous aggregation of alcohol dehydrogenase [44]. An equimolar concentration of βS achieved 50% suppression of temperature-induced amorphous aggregation of citrate synthase. Finally, an equimolar concentration of βS inhibited the amyloid fibrillar aggregation of amyloid β_1–40_ [44], although one study reports that βS further enhanced the aggregation of this Alzheimer’s disease (AD)-related peptide [45]. Thus, βS is not a specific molecular chaperone to αS, whereby it possesses mechanisms to prevent both amorphous and fibrillar protein aggregation. Similarly, αS possesses in vitro chaperone ability against a diversity of stressed proteins [46].

### 4.2. β-Synuclein Regulates Synaptic Function, Lipid Binding, and Dopamine Neurotransmission

βS contains five highly conserved, amphipathic KTKEGV sequences in its N-terminal region (Figure 1), which have significant sequence similarity to class A2 apolipoproteins, suggesting a lipid-binding role of βS, particularly via its N-terminal region [47,48]. Furthermore, lipids have been implicated in their involvement with PD and dementia with Lewy bodies (DLB), whereby pathogenic brain homogenates have a high level of polysaturated fatty acids [49], with Lewy bodies containing a particularly large amount of lipids [50]. In vivo, βS is associated with fatty acids. For example, the detection of βS was enhanced when delipidation was performed on mouse brain homogenate [51] and when DLB brain homogenates were processed with a hydroxyalkoxypropyl-dextran Lipidex-1000 fatty acid binding column [52]. Finally, βS displayed a higher affinity for smaller liposomes, which mimic the size of cellular transport vesicles, suggesting an intrinsic role for βS in lipid binding [53]. Lipid binding of the synucleins reflects functional compensation, whereby in the presence of αS, the membrane association of βS and γS was enhanced. Concomitantly, the presence of βS and γS attenuated the membrane association of αS [54]. Taken together, these studies suggest that lipid binding is an evolutionary conserved biological role of βS.

#### 4.2.1. Structural Changes to β-Synuclein upon Lipid Binding

βS is intrinsically disordered, but upon binding to surfactants and lipids, it rearranges into a predominantly α-helical protein [26,55,56,57]. NMR spectroscopy revealed that the binding of βS to the anionic detergent, SDS, induced a high content of α-helicity in its entire N-terminal region [55]. Another study reported varying degrees of α-helicity in the N-terminal and NAC regions of micelle-bound βS, in which a lower degree of helicity was observed for H65-N76, along with a break in helicity at K43-T44 [56]. Strong NMR ^1^H-^1^H nuclear Overhauser effects suggested that H65-E83 in the NAC region, populates both an α-helical and unstructured conformation upon binding to lipids [56]. The structure of βS was predicted using AlphaFold [58] (Figure 2). The structure predicts an α-helical tendency for the N-terminal and NAC regions of βS, consistent with the adoption of such conformation for these regions in the presence of lipids, with an unstructured C-terminal region. Indeed, in the presence of SDS, the C-terminal region of βS remains largely unperturbed and unstructured, consistent with its lack of involvement in lipid binding [55]. In agreement with this conclusion, C-terminal truncation of βS did not significantly alter its binding affinity to 1,2-dimyristoyl-sn-glyerco-3-phospho-l-serine (DMPS) vesicles in comparison to the full-length protein [53]. The C-terminal-truncated and the full-length protein had large differences in overall charge of −1.2 and −14.8, respectively [53], implying that electrostatic interactions may not be the predominant effects in regulating the interaction between βS and lipids. This notion is supported by the ability of full-length βS to bind to 100% zwitterionic liposomes with a higher affinity than predicted based on its amino acid sequence [53]. On the contrary, βS had a greater affinity to liposomes with higher anionic content [53]. The addition of cholesterol, which reduces the permeability and flexibility of the membrane, did not change the ability of βS to bind to the membrane [53]. These findings suggest that βS employs a plethora of electrostatic and other types of interactions to bind to lipids, which mainly occurs via the N-terminal region, and possibly part of the NAC region.

Lipid binding has also been implicated in the oligomerization and fibril formation of αS, however, whether this is applicable to βS remains contentious. For example, βS failed to form amyloid fibrils upon binding to DMPS vesicles at the same concentration used to induce fibril formation of αS [26]. Similarly, when polyunsaturated fatty acids were added to the extracellular media of primary mesencephalic neurons, intracellular oligomerization of αS was enhanced, but not of βS, suggesting that lipid binding does not promote the aggregation of βS. On the contrary, when βS was incubated with a 3.5–11 molar excess of SDS in vitro, βS formed Thioflavin T and Congo red positive amyloid fibrils which were confirmed by electron microscopy. Similarly, the incubation of βS with a high concentration of α-linoleic acid enhanced its oligomerization [51], suggesting that lipid binding induces the aggregation of βS. In summary, there are secondary structural changes of βS upon lipid binding to adopt a more structured (α-helical) conformation. However, the type of lipid and experimental conditions modify the aggregation behavior of βS.

#### 4.2.2. Structural Changes to the Lipid upon Binding of β-Synuclein

Lipid binding induces a structural change in βS, but βS can also alter the structure of the lipid. The incubation of βS with a 20-molar excess of 1-palmitoyl-2-oleoyl-phosphatidylglycerol (POPG) vesicles resulted in the vesicle rearranging into a tubular structure, with higher concentrations of βS promoting a more curved structure. Considering that βS has five amphipathic lipid binding repeats in its N-terminal and NAC regions compared to six in αS, a higher concentration of βS was required to achieve the same degree of vesicle curvature induced by αS [57]. The change in membrane curvature induced by βS led to significant vesicle leakage, with αS inducing slightly more disruption to the vesicle integrity in comparison to βS [57]. On the contrary, a separate study reported that monomeric and protofibrillar βS were unable to tightly bind to phosphatidylglycerol lipid membranes, with no changes in its membrane integrity observed [59]. Despite these conflicting lines of evidence, mice that had knocked out αS, βS, and γS had an increased level of endophilin A1, endophilin B2, synapsin II isoform b, and annexin A5, which all possess lipid-binding activity and physiologically generate membrane curvature [48]. The upregulation of these proteins may indicate a functional compensatory role for the synucleins. Taken together, these results demonstrate that βS binds to lipids which may induce a change in the overall structure or morphology of the lipid.

#### 4.2.3. β-Synuclein Regulates the Nigrostriatal Dopaminergic System

In PD, dopamine (DA)-producing neurons, which project from the substantia nigra to the dorsal striatum, are particularly vulnerable to degradation. DA is a neurotransmitter which regulates motor function. It is susceptible to oxidation at physiological pH, leading to the formation of highly reactive intermediates such as aminochromes which cross-link and/or inactive proteins [60,61]. Furthermore, oxidized DA has been implicated in mitochondrial and lysosomal dysfunction in PD [62]. Hence, DA is sequestered into acidic synaptic vesicles via the vesicular monoamine transporter-2 (VMAT-2). Recent studies have shown that mice which lack βS exhibit diminished VMAT-2-dependent DA uptake [63], suggesting a role of βS in DA sequestration. In addition, the active metabolite of 1-methyl-4-phenyl-1,2,3,6-tetrahydropyridine (MPTP), 1-methyl-4-phenylpyridinium (MPP+), is commonly used to induce a Parkinsonian phenotype in mouse models via injection into the dopaminergic neurons of the substantia nigra. For MPP+ to be toxic, it enters the neuron via the DA transporter (DAT) rather than being sequestered into synaptic vesicles via VMAT-2. Chemically, MPP+ is similar in structure to DA. Dopaminergic neurons of the substantia nigra are particularly susceptible to MPP+ because they have a higher ratio of DAT:VMAT-2 in comparison to other neurons of the brain. Mice expressing αS and γS are singularly or doubly susceptible to MPP+ toxicity, with the lack of these two proteins conferring resistance. However, the absence of all three synucleins led to mice that were equally susceptible to MPP+ as their WT counterparts, suggesting a role of βS in their resistance [63]. Indeed, mice lacking βS singularly or in combination with αS or γS exhibited a similar degree of susceptibility to MPP+ toxicity as their WT counterparts, highlighting the key role of βS to confer resistance, likely via its sequestration through VMAT-2-related mechanisms. Thus, it seems that βS can rescue dopaminergic neurons from the MPP+ toxicity although its interaction with, or the presence of, other synucleins appears to play a role. Analysis of synuclein-null synaptic vesicles with restored βS expression showed an increase in DA uptake and the presence of different proteins in comparison to their controls. Of interest, the restoration of βS upregulated tyrosine hydroxylase (TH) and aromatic l-amino decarboxylase (AADC), in which both proteins regulate DA synthesis and form transient complexes with VMAT-2. Overall, these results indicate the novel role of βS to improve DA uptake by initiating the formation of a complex between AADC, TH, and VMAT-2, which may confer an allosteric effect on the DAT [63]. A separate study showed that aged mice lacking βS exhibited diminished coordination, grip strength, sensorimotor function, and endurance in comparison to mice expressing βS, suggesting that βS is important to regulate motor performance [9]. Moreover, aged mice which lacked all three synucleins showed decreased DA levels and increased DA turnover in the dorsal striatum [9]. Thus, βS is a crucial protein which regulates DA in nigrostriatal neurons and concomitant motor function.

### 4.3. β-Synuclein Regulates Cellular Metal Levels

Due to the lack of structure and inherent dynamism, IDPs can interact with and stabilize metal ions, suggesting the presence of metal binding modes on these proteins [64]. Imbalances in metal homeostasis have been reported in neurodegenerative diseases. For example, in PD, high levels of zinc and copper are present in the brain and cerebrospinal fluid (CSF) [65], and epidemiological studies reported an increased risk of developing PD upon environmental heavy metal exposure [66]. In AD, high levels of copper, iron, and zinc have been found in amyloid plaques [67]. These studies illustrate a relationship between neurodegenerative diseases and metals.

Due to the implication of the synucleins in neurodegenerative diseases, the metal-binding capability of αS and βS has been investigated. αS forms cytotoxic oligomers upon its interaction with copper. On the contrary, βS did not significantly affect the viability of human neuroblastoma SH-SY5Y cells upon interaction with heavy metals [68], despite the binding of βS to these metals. Primarily, βS binds Cu(II) at M1 with high affinity by forming a stable coordination sphere, which also involves the amide nitrogen and carboxylate oxygens of D2. The reducing environment intracellularly would suggest that Cu(I) binding to βS is more likely, in which βS binds via the thioether group of M1. A second, high-affinity site is present at H65, in which Cu(II) binds through its imidazole ring. A third biding site with low affinity was identified in the C-terminal region, with the carboxylate groups of the D121 and E126 predominantly responsible for binding [69]. The metal binding residues of βS are indicated in Figure 1. As these copper binding sites of βS are similar to those of αS, it is suggested that βS may play a vital role to chelate copper ions, thus removing free, redox-active Cu(I), which can produce free radicals and promote the formation of toxic αS oligomers via the presence of redox-active Cu(I) [69]. In addition, the overexpression of βS in a human neuroblastoma cell line reduced cellular iron levels, which may also reduce neurotoxicity [35,70]. Despite this, high concentrations (i.e., mM in comparison to the μM range used for the previously described studies) of copper and other heavy metals can induce fibril formation of βS [71]. Finally, independent of αS and γS, the expression of βS is regulated by metal transcription factor-1 (MTF-1) [72]. An overexpression of MTF-1 resulted in a 30-fold increase in βS levels in a human neuroblastoma cell line. As MTF-1 promotes the expression of heavy metal-buffering metallothionein proteins, the cell may regulate the expression of βS to also chelate metals and reduce cellular toxicity [72]. In summary, it is apparent that βS binds metals to regulate cellular metal homeostasis.

### 4.4. β-Synuclein Regulates Apoptosis

It has been hypothesized that βS is neuroprotective by protecting cells against apoptosis. Firstly, telencephalon-specific murine (TSM1) neurons stably expressing βS were treated with staurosporine, a compound which induces apoptosis via the activation of caspase-3 [73]. A significant decrease in the number of cells that possessed fragmented DNA resulted, as monitored by terminal deoxynucleotidyl transferase dUTP nick-end labelling (TUNEL) assays, suggesting that apoptosis was inhibited by βS [73]. Similarly, when βS-expressing TSM1 cells were treated with the dopaminergic toxin 6OHDA, the activation of caspase-3 was inhibited, further corroborating that βS is anti-apoptotic [73]. A separate report demonstrated that the addition of 1 ng/mL of recombinant βS, a concentration that reflects its physiological levels in the CSF, to the extracellular media of brain microvascular endothelial cells led to a decrease in the number of TUNEL-positive cells, suggesting that βS was neuroprotective. On the contrary, the addition of 50 ng/mL and 500 ng/mL of βS, reflective of levels in neurodegenerative states, significantly increased the number of TUNEL-positive cells, suggestive of cellular death through apoptosis [74]. Overall, these findings imply that βS is protective against apoptosis at low, physiological concentrations, but high levels lead to cellular toxicity.

At physiological concentrations, the anti-apoptotic phenotype of βS may be p53-dependent. For example, the stable expression of βS showed decreased levels of cellular p53 and phosphorylated p38, a protein which activates p53 by phosphorylation, and increased levels of Mdm2, a protein which controls the expression of p53 by regulating its ubiquitination and degradation rates. As p53 expression is regulated by these proteins, the change in their expression levels suggests that the anti-apoptotic behavior of βS is p53-dependent [73]. A separate study reported an increase in the nuclear localization of p53 and stronger nuclear and cytoplasmic staining of Mdm2 upon the addition of βS to the extracellular media of brain microvascular endothelial cells [74]. The extent of p53 and Mdm2 upregulation did not differ between the amount of βS added. However, there was a decrease and increase in the rate of apoptosis upon treatment with βS at concentrations reflecting physiological and neurodegenerative states, respectively [74]. These findings suggest that p53-independent mechanisms contribute to the apoptotic roles of βS.

The serine threonine kinase, Akt, has also been proposed to explain the anti-apoptotic effect of βS. Akt inhibits apoptosis by phosphorylating Mdm2, which is then translocated to the nucleus to bind to p53. B103 rat neuroblastoma cells overexpressing βS conferred resistance to cell death induced by rotenone, a mitochondrial complex I inhibitor, but the downregulation of Akt abolished resistance to its neurotoxic effects [27]. Moreover, the pathways of βS and Akt to inhibit apoptosis appear related, as the transient overexpression of βS led to an upregulation of Akt activity and enhanced neuroprotection against rotenone. Concomitantly, a downregulation of βS decreased Akt activity, and cells were more susceptible to the toxic effects of rotenone. Co-immunoprecipitation of Akt and βS from rat B103 neuroblastoma cells was also observed, suggesting a direct interaction between these two proteins [27]. However, a separate study reported a decrease in Akt immunoreactivity when brain microvascular endothelial cells were treated with 1, 50, and 500 ng/mL of βS [74]. Taken together, although whether the apoptotic function of βS is dependent on Akt remains contentious, βS appears to be involved in regulating apoptosis.

### 4.5. β-Synuclein Regulates Protein Degradation Pathways

βS is both directly and indirectly implicated with the autophagy-lysosomal pathway, a complex network of cellular machinery and processes which degrade damaged or unwanted intracellular macromolecules, including protein aggregates [75]. For example, in the frontal cortex of DLB brains which had elevated levels of βS, there was an increase in the selective autophagy-lysosomal pathway marker SQSTM1/p62, which co-localized with βS [76]. In addition, DLB brains showed a distorted staining pattern of SCARB2/Limp2, a lysosomal marker in which βS was consistently present within Limp2-positive vacuoles. Moreover, DLB brains exhibited an increase in a selective autophagosome marker, LC3-II. HeLA and human neuroblastoma BE(2)-M17 cells overexpressing βS showed increased levels of LC3-II and its puncta. These cells also had diminished ability to turnover LC3-II, suggesting that its autophagic degradation activity or autophagy flux was weakened [76]. A separate study demonstrated that treatment of yeast cells expressing βS with PMSF, an inhibitor of the autophagy/vacuolar pathway, caused an increase in the number of cells containing aggregates, implying that this pathway is involved in the clearance of βS aggregates in yeast cells [77]. Overall, it is concluded that βS can influence the autophagy-lysosomal pathway.

Recently, βS has been observed to form amyloid fibrils under mildly acidic conditions, which may resemble the environment of cellular compartments such as the endocytic pathway, late endosomes and cytosolic acidification as a result of oxidative stress [78]. Glutamic acid residues at positions 31 and 61 of βS (Figure 1) were responsible for promoting aggregation upon their protonation at low pH [78]. Although AFM revealed that these βS amyloid fibrils formed under acidic conditions structurally resembled those formed by αS, their effect on cellular toxicity has not been investigated [78]. Thus, despite βS preventing the aggregation of αS, βS can also form amyloid fibrils, which may lead to dysfunction in acidic cellular microenvironments such as within the autophagy-lysosomal pathway.

Finally, there is evidence implicating βS with the ubiquitin-proteasome pathway, an ATP-dependent process which degrades damaged proteins. When yeast cells expressing βS were treated with a proteasome inhibitor of the ubiquitin-proteasome system (UPS), MG132, there was a two-fold increase in cells with βS aggregates [77], suggesting that the UPS is involved in clearing βS aggregates in yeast cells.

### 4.6. β-Synuclein Promotes Cellular Toxicity and Protein Aggregation

#### 4.6.1. Changes in β-Synuclein Expression in Pathology

Pathological models and tissues have reported changes in the expression of βS, suggesting that βS is involved in disease processes. Biological samples isolated from neurodegenerative, neurodevelopmental, neuroinflammatory disease, or neuro-cancerous tissue show an increase in βS protein expression, a decrease in βS mRNA, and no significant change in genetic expression. For example, in the hippocampus of PD and DLB patients, βS was immunopositive in the axonal terminals [79]. In the frontal and occipital cortices of DLB patients, there was an a upregulation and downregulation of βS, respectively [76]. A high level of βS was also observed in the CSF of patients with DLB, AD, and Creutzfeldt-Jakob disease [80]. In multiple system atrophy, βS was extensively present in Purkinje cells [81]. In AD white matter, there was a 4.0- to 5.6-fold increase in βS [82]. Furthermore, Hallervorden-Spatz disease or brain iron accumulation type 1, a rare neurodegenerative disorder sharing similar pathology to PD and DLB, reported βS-positive spheroids in brain tissue [83]. Zebrafish with downregulated PLAG2G6, a gene encoding for a Ca^2+^-independent phospholipase A2 group 6, in which mutations in this gene result in neuroaxonal dystrophy, neurodegeneration with brain iron accumulation and juvenile Parkinsonism, showed an increase in βS expression [84]. Patients with multiple sclerosis (MS) also had a 2.5-fold higher expression level of βS mRNA [85]. An accumulation of βS was also observed in Sandhoff disease, a neurodegenerative disorder associated with defective lysosomal enzymes and changes in lysosomal storage [86,87]. Children with autism spectrum disorder had significantly higher levels of plasma βS [88]. βS was also present in ependymal tumors, pilocytic astrocytomas, glioblastomas, anaplastic oligodendrogliomas, and primitive neuroectodermal tumors/medulloblastomas [89]. On the contrary, a downregulation of βS mRNA in diffuse Lewy body disease (DLBD), AD [90], and in the lymphocytes of patients with schizophrenia [91] has been reported. Gene variant testing revealed that βS was not a susceptibility gene for PD [92] and that there were no significant association of the βS gene to DLBD [93]. Thus, disease states reflect modified βS expression. However, these conclusions must be interpreted with caution, as βS is not present in Lewy bodies or glial cytoplasmic inclusions [81], and the expression of βS may be inversely correlated with that of αS (i.e., an increase in βS leads to a decrease in αS and vice versa [74]), suggesting that it may be difficult to draw conclusions on whether these changes in βS levels are a result of, or contribute to, disease progression directly or via αS.

#### 4.6.2. β-Synuclein Can Induce Neurotoxicity

Despite the plethora of evidence that βS is neuroprotective, some studies suggest that βS is neurotoxic. When primary cortical neurons were transduced with AAV-6 vectors expressing βS-EGFP, cell death occurred after 10 to 14 days, albeit to a lower extent in comparison to cells expressing αS-EGFP [94]. βS-EGFP-expressing cells were also metabolically impaired, indicated by the lack of increase in EGFP fluorescence, which was observed in EGFP-only transfected cells [94]. Interestingly, when αS was expressed with a 10 or 30 molar excess of βS, cell death was partially rescued, suggesting that dysregulation in the interplay of αS and βS may contribute to the toxicity of βS. Moreover, when AAV-2 vectors expressing βS were injected into the substantia nigra of mice brains, 20% of dopaminergic neurons were lost two weeks post-injection. At the same time point, 46% of neurons were lost in mice injected with αS [94]. βS-expressing mice required eight weeks to achieve 45% of neuronal loss [94].The immunohistochemical analysis of these brains revealed a significant amount of βS aggregates, which were amyloid fibrillar in form [94]. Moreover AAV-transduced primary neurons expressing βS contained destroyed fragmented mitochondria after 13 days of expression. Despite this, the remaining mitochondria could still actively remove elevated Ca^2+^ levels from their matrix and displayed no significant impairment in their motility, membrane potential or energy production [94]. These results suggest that βS is neurotoxic but not to the extent of αS, with βS exhibiting toxicity through more than one pathway.

The expression of βS in yeast cells demonstrated that βS is toxic and induces aggregate formation in a similar manner to αS. Spotting assays revealed that yeast cells expressing βS had reduced growth in comparison to cells that were transfected with an empty vector [77,95]. Moreover, 16% of βS-expressing yeast cells were propidium iodine positive in comparison to 4% of empty vector transfected cells, and 20% of αS-expressing cells, suggesting that βS expression was cytotoxic, although to a lesser extent than αS [77,95]. βS-expressing yeast cells also displayed intracellular inclusions in which fluorescence recovery after photobleaching revealed that they had comparable protein-immobilized fractions and mean residence times as the inclusions present in αS-expressing yeast cells [95]. Moreover, βS-expressing yeast cells also showed a significant increase in the levels of superoxide radicals, which was also observed in αS-expressing yeast cells. Contrary to previous studies, the co-expression of βS with αS exacerbated cytotoxicity [95]. Thus, βS can induce cytotoxicity and aggregate formation in yeast cells.

In addition, βS plays a key role in neuroinflammatory autoimmune diseases such as MS. Rats injected with recombinant βS with prior treatment with cyclophosphamide, an immunosuppressant and hence an enhancer of the immune response, developed autoimmune encephalomyelitis, a model of human autoimmune disease [96]. Upon further investigation of the specific epitope to induce the inflammatory response, L93-L111 of βS was selected based on its ability to bind to the MHC class II I-A binding motif. Indeed, injection of this βS peptide into mice resulted in symptoms characteristic of autoimmune encephalomyelitis and uveitis [17]. When T cells which specifically recognize L93-L111 of βS were generated and injected into mice, they developed the same neuroinflammatory symptoms. In a separate study, D92-P110, V84-I108, and P96-E120 of βS also induced experimental autoimmune encephalomyelitis in mice upon prior treatment with cyclophosphamide [96]. Despite P96-I108 being common to these three peptides, it did not induce experimental autoimmune encephalomyelitis in mice. Thus, βS induces neuroinflammation and experimental autoimmune encephalitis, which may be mediated by T cells [17].

Recently, it has been observed that βS-specific T cells induce neuroinflammation and neurodegeneration in the gray matter of the brain, which is a key process in mediating brain atrophy and disease progression in MS. When rats received βS-specific T cells via intravenous injection, the cells migrated to the gray matter of the brain. As a result, rats presented with equivalent symptoms of MS such as paresis of the individual limbs, head tilting, and ataxia. Moreover, histological analysis revealed that βS-specific T cells were in close association with neurons, with some already breaching into the neuron, suggestive of neuronal damage. The gray matter of these brains contained a significant number of activated glia, apoptotic neurons, and a reduction in the number of synaptic spines, suggestive of neuronal inflammation and degeneration induced by βS-specific T cells. Despite the neuroinflammatory response subsiding shortly after, repeated bouts of neuronal insult via multiple βS-specific T cell injections led to significant gliosis, neuronal damage, atrophy of cortical tissue, and permanent damage of the brain matter. Patients with chronic progressive MS had significantly higher amounts of βS-specific T cells in their blood. They also had a higher levels of αS-specific T cells. The blood of PD patients had higher levels of αS- but not βS-specific T cells [97]. A previous study also reported that immunization of rats with recombinant βS, but not αS, induced experimental autoimmune encephalitis [96]. Overall, although βS is probably involved in mediating neurodegeneration and neuroinflammation in MS, its actions may not directly underlie PD-related neurotoxicity.

## 5. Pathological Mutations of β-Synuclein

Two missense mutations of βS have been linked to disease, a proline to histidine substitution at residue 123 (P123H) in the C-terminal region, and a valine to methionine substitution at residue 70 (V70M) in the NAC region. Thus, the substitution of a sterically hindered proline, an amino acid that promotes protein disorder, by a hydrophilic histidine and conversion of a branched valine to a linear methionine with a large sulfur group lead to subtle conformational changes in βS that promote a pathological phenotype to βS.

The autosomal dominant P123H βS mutation was first discovered by Ohtake et al. in 2004 in one proband DLB patient [98]. In P123H βS transgenic mice, a histopathological analysis revealed an accumulation of insoluble P123H βS and axonal swellings in the striatum and globus pallidus, which increased in an age-dependent manner. Axonal swellings with globules contained LC3-II, an autophagosome marker, and were immunopositive for αS and ubiquitin, suggesting abnormal protein accumulation due to compromised lysosomal degradation. Globules were found in GABAergic neurons which lacked synaptic and axonal markers such as synaptophysin, synapsin, and neurofilaments, implying that these neurons were non-functional. Outside the globules, there was an upregulation of heat-shock proteins Hsp70 and the sHsp, Hsp25, at six months of age and a substantial decrease in synaptic markers SNAP25 and VAMP5 at 18 months. Lewy body-like inclusions were not present in P123H βS transgenic mice, which accurately reflects the lack of βS in human Lewy bodies. The mice were accompanied by substantial gliosis and an increase in an astrocyte activation marker, GFAP. Behavioral analysis of P123H βS mice showed significant learning and memory deficits assessed through the water maze and target quadrant tests, and a decrease in spontaneous activities. No motor deficits were observed at six months, but were present in the later stages [99]. In a separate study, in comparison to WT βS mice, P123H βS mice had a significant decrease in mRNA levels of mature cell markers such as *Tdo2* and *Dsp*, which also occurs in some neuropsychiatric disorders and AD mouse models. An increase in the immunoreactivity of ionized calcium-binding adaptor molecule 1 was also observed, suggesting activated microglia and neuroinflammation in P123H βS mice [100]. In summary, P123H βS exhibits neurotoxic gain-of-function.

Moreover, P123H βS disturbs the balance between βS and αS and promotes overall neurotoxicity. For example, crossing P123H βS transgenic mice with αS transgenic mice exacerbated neurodegeneration. The double transgenic mice had higher mortality, a lower body weight, and increased aggregation of both αS and P123H βS. Additionally, there was a decrease in the expression of synaptic proteins such as VAMP2, SNAP, and PSD95 and a significant upregulation of pro-inflammatory factors, suggesting the presence of neuroinflammation. Neurodegeneration in double transgenic mice was not attributed to apoptosis due to TUNEL-negative cells, but the degeneration of dark neurons in the hippocampus and cortex was observed, as is present in pathological conditions such as ischemia, epilepsy, and head injury. Furthermore, double transgenic mice exhibited motor dysfunction such as impairment of locomotor performance and clasping behavior attributed to the degeneration of the nigrostriatal dopaminergic neurons. A concomitant decrease in the concentration of DA and TH was observed [99]. At the molecular level, NMR studies have shown that P123H βS disrupts the extended PP-II motif in the C-terminal region [101]. The P123H mutation is located in a region of βS (P114-A134) which, as discussed above, interacts with and prevents αS aggregation [28,36,101]. In parallel, this mutant promoted the aggregation of αS in vitro [101]. Hence, P123H βS negatively alters the interactions of βS with αS, leading to cytotoxicity.

Furthermore, cell-based studies of P123H βS have provided detail on the mechanisms contributing to its neurodegenerative phenotype. For example, B103 neuroblastoma cells showed large lysosomal inclusions which were cystic membrane and electron-dense myelinosome-like, as is observed in ganglioside-related lysosomal storage disorders. Accordingly, these inclusions were immunopositive for lysosomal markers such as cathepsin B, LAMP-2, GM2, GM1, with partial detection of the aggresomal marker, γ-tubulin. ATP13A2, a lysosomal type 5p-type ATPase associated with early onset PD, co-localized with P123H βS in lysosomal inclusions [102]. P123H βS also showed abnormal lipid binding, in which it had a five times higher affinity for liposomes made from equal components of a zwitterionic and an anionic lipid in comparison to WT βS [53]. Hence, lysosomal and lipid abnormalities may underlie the toxicity of P123H βS.

The V70M mutation was also discovered by Ohtake et al. in an 83-year-old Japanese man with sporadic DLB [98]. Cell culture models expressing V70M βS were employed to investigate the molecular mechanisms underlying its pathology. V70M βS-transfected B103 rat neuroblastoma cells showed occasional cytoplasmic aggregation and inclusions positive for lysosomal markers such as cathepsin B, LAMP-2, and ganglioside GM2, suggesting that this mutation leads to lysosomal dysfunction [102]. Moreover, the expression of V70M βS in yeast cells led to an approximately two-fold increase in inclusion formation in comparison to WT βS [95]. Interestingly, these inclusions were not toxic [95]. These results suggest that V70M βS promotes aggregate formation and exerts toxicity through lysosomal dysfunction.

In conclusion, both the P123H and V70M βS encourage neurotoxicity through lysosomal dysfunction. The former mutant additionally exerts toxicity by destabilizing αS-βS interactions, calcium homeostasis, and/or abnormal lipid binding.

## 6. Post-Translational Modifications of β-Synuclein

As is characteristic of other IDPs, βS employs PTMs to fine-tune its regulatory interactions [1]. For example, βS is modified by β-*N*-acetylglucosamine linked to hydroxyl groups on the sidechains of serine and threonine residues via the activity of O-GlcNAc transferase and *N*-acetyl-β-d-glucosaminidase, enzymes which are abundant in the synaptosome cytosol [103]. Although the exact function of this βS modification is unknown, proteins with attached O-GlcNAc are implicated in signal transduction and neurodegenerative diseases, with mutations in this gene responsible for cell viability and development. Interestingly, this PTM is present in βS but not αS. Proteomics revealed that βS is also modified by protein l-isoaspartate O-methyltransferase (PIMT), leading to the methylation of isoaspartic acid residues of βS. Isoaspartic acid residues in proteins arise from spontaneous, often age-related, non-enzymatic modification to aspartic acid and asparagine residues. Methylation via PIMT facilitates the restoration of these residues in the damaged proteins [104,105]. Stable oligomers of αS also contained methylated isoaspartatic acid residues. Interestingly, in conditions which mimic ageing, αS was modified 20 times more rapidly than βS, with βS accumulating only low levels of this PTM. When αS and βS were co-incubated, there was a significant reduction in the methylation of isoaspartic acid residues of αS by PIMT, suggesting a regulatory role of βS on αS [105]. When expressed in yeast, all three lysines in βS are conjugated with the small ubiquitin-like modifier (SUMO), which has been linked to a cytoprotective phenotype [77].

Phosphorylation is a common PTM of βS. For example, S118 of βS is phosphorylated by human polo-like kinase 1 and 3, enzymes which regulate the cell cycle, cellular stress responses and carcinogenesis [106]. The phosphorylation of S118 occurs in the C-terminal region of βS, in a comparable position of the amino acid sequence to the phosphorylation of S129 in αS (Figure 1). Phosphorylation of S129 in αS is a marker for PD and related diseases [106]. Similarly, G-protein receptor kinase-2 preferentially phosphorylates both αS and βS [107] and thereby inhibits their interaction with phospholipids and phospholipase D2 [107,108], an enzyme which regulates phosphatidylcholine breakdown and regulates vesicular trafficking. Both observations are consistent with the lipid-binding function of αS and βS. Serine residues were also found to be phosphorylated by calmodulin-dependent protein kinase II in vitro [109]. In addition, Y127 of βS expressed in the rat brain cytosol was phosphorylated by the tumor suppressor enzyme, Src kinase-homologous kinase [110]. On the contrary, βS can also stimulate the dephosphorylating enzyme, serine/threonine protein phosphatase 2Ac (PP2Ac), a subunit of the PP2A holoenzyme. In neurodegenerative diseases such as DLB, there is a decrease in the activity of PP2Ac [111]. Taken together, the exact function of each PTM on βS function is not known, with most appearing to be involved in key regulatory processes of the cell, with PTM dysfunction being implicated in neurodegenerative diseases.

## 7. Future Directions and Conclusions

As an IDP, βS is unstructured and hence is malleable to interact with a variety of macromolecules and ligands under different cellular conditions. The current review has discussed the structure and function of βS, focusing on its ability to prevent protein aggregation, regulate synaptic function and lipid binding, mediate apoptosis, participate in protein degradation pathways and promote cellular toxicity. The effects of pathological mutations and PTMs of βS were also explored. The diverse function roles of βS are summarized schematically in Figure 3. Overall, βS has either a synergistic or antagonist relationship to αS but also possesses functions independent of αS.

Most of the research examining βS has resulted in it being deemed a partner protein of αS, diminishing its importance and discouraging further investigation. Considering the recent discoveries attributing novel roles to βS, as outlined above, more work is warranted to understand better the function of βS, especially its relationship to neurological and neurodegenerative diseases. In the following, a few experiments are proposed to address these gaps. Firstly, obtaining high-resolution structure(s) of βS is desirable. In the Protein Data Bank (PDB), 51 structures of αS have been deposited (e.g., peptide fragments, complexes, mutants, and fibrils, etc., under various experimental conditions). However, no structures of βS are present in the PDB. Recent computational algorithms to predict protein structure such as AlphaFold [58] have improved accuracy, but the structure prediction of IDPs and aggregates based on amino acid sequences is challenging. When AlphaFold was used to determine the structure of βS (Figure 2), approximately 77% of the protein was predicted with low or very low confidence. Experimental techniques such as cryo-EM or solid-state NMR spectroscopy could be applied to obtain the structure of higher order and lipid-bound βS, as well as the aggregates formed by P123H βS, V70M βS, and the fibrillar structures of WT βS induced by mildly acidic conditions. Secondly, through its cellular protein interactors, novel biological processes or molecular functions of βS could be uncovered. The Biological General Repository for Interaction Datasets (BioGRID) [112] lists 254 and 16 known human protein interactors of αS and βS, respectively. Employing in-cell proximity labelling technologies which capture transient interactions [113], i.e., those characteristic of IDPs, could unveil novel interactors and in turn provide a better understanding of βS function within the cell. Thirdly, PTMs are characteristic of IDPs such as βS, which thereby confers additional functionality. High resolution mass spectrometry coupled with extended modification searching of βS isolated from various basal and pathological in vivo and in vitro model systems may provide further insight into the functional role of βS in cells.

A deeper understanding of βS biology and chemistry may confer this protein as a potential candidate as a biomarker and/or for use therapeutically as a treatment for synucleinopathies [33,42,43]. However, the multifaceted nature of these diseases, the widespread expression of βS within the brain, its intrinsically disordered nature, the differences between the transcriptomic and proteomic expression profiles and the functional compensation between the synucleins, and the potentially varied and large repertoire of protein interactions dependent on the cell’s dynamic needs may make these endeavors challenging. Nonetheless, further investigation of βS is warranted to identify its role and relevance in cellular homeostasis and pathogenesis.

In conclusion, the intrinsically disordered nature of βS underpins its ability to regulate a plethora of important cellular processes including the prevention of αS aggregation and its involvement in a variety of neurological diseases.

## Figures and Tables

**Figure 1 biomolecules-12-00142-f001:**
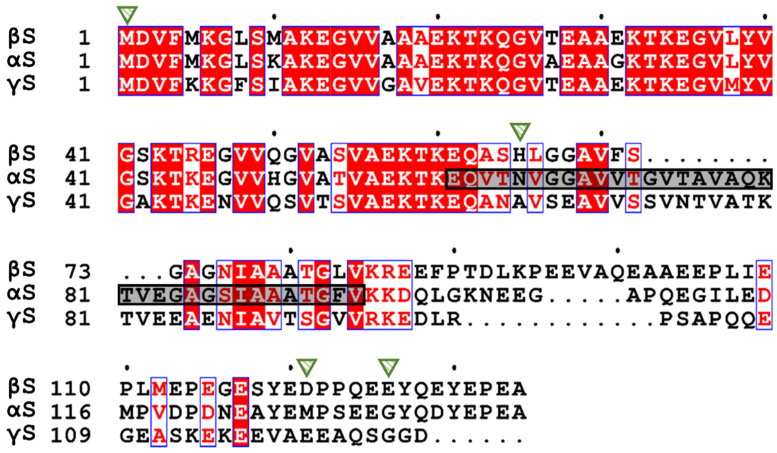
Amino acid sequence alignment of human α-, β-, and γ-synucleins. The one-letter amino acid code in the red shaded boxes with white text and the blue outlined boxes with red text represent 100% sequence similarity and amino acids with similar chemical properties, respectively. The black dots are placed at every 10 amino acid intervals of βS. The inverted green triangles represent the metal binding residues of βS. The gray shaded box with the black outline in the αS sequence represents the NAC region (E61-V95) in αS. M1-K60 in all three proteins encompass the N-terminal region, and K85-A134 refers to the C-terminal region of βS. The multiple sequence alignment was performed on the Network Protein Sequence @nalysis server using CLUSTAL W (1.8) [12] and visualized using Easy Sequencing in PostScript ESPript 3.0 [13].

**Figure 2 biomolecules-12-00142-f002:**
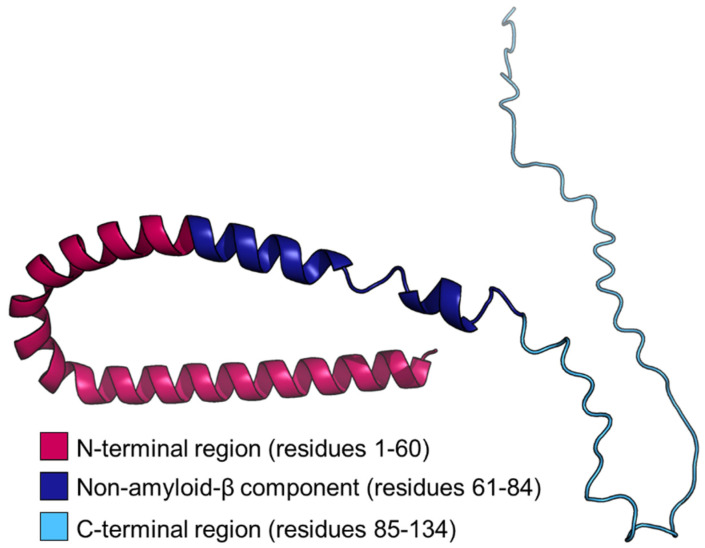
Predicted structure of βS. The sequence of human βS (UniProt accession code Q16143) was used to predict its full-length protein structure using AlphaFold [58] and visualized using Pymol. βS is arranged in a tripartite arrangement, with the N-terminal region (M1-K60, pink), central NAC region (E61-V84, dark blue), and the C-terminal region (K85-A134, light blue). The predicted N-terminal helical structure is reflective of the lipid-bound conformation of βS in this region. Model confidence was quantified with per-residue confidence scores (pLDDT) generated by AlphaFold, ranging from 0 to 100. Residues 1–30 and 32 were predicted with confidence with pLDDT scores of 70 to 90. Residues 31, 33–87, 91, 109, 113–114, 119–123, 125, and 127–133 were predicted with low confidence, i.e., with pLDDT scores of 50 to 70. Residues 88–90, 92–108, 110–112, 115–118, 124, 126, and 134 were predicted with very low confidence, i.e., with pLDDT scores below 50. Scores below 50 may suggest an unstructured region. No residues in human βS were predicted with very high confidence, i.e., with pLDDT scores above 90.

**Figure 3 biomolecules-12-00142-f003:**
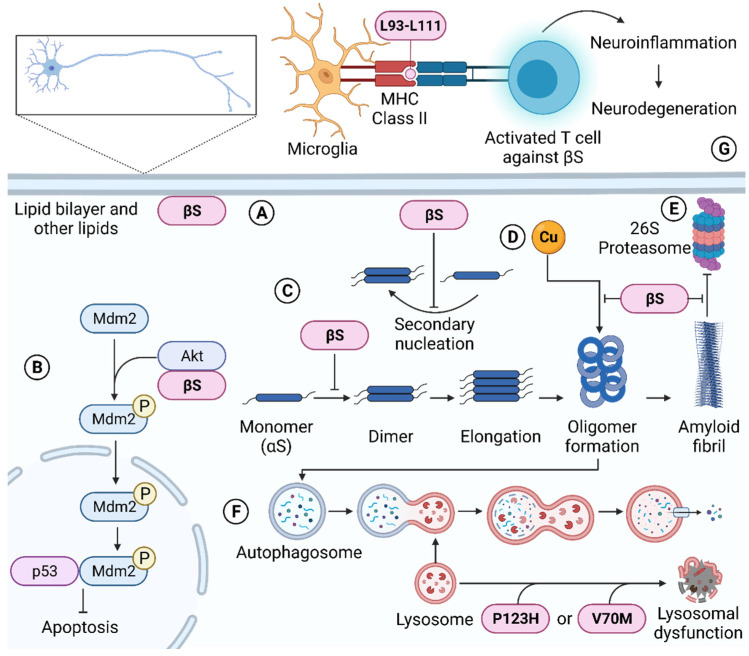
The diverse functions of βS. Inside the neuron, βS binds to lipids through its N-terminal region (**A**). βS also inhibits apoptosis (**B**), potentially via its direct interaction with Akt. Akt phosphorylates Mdm2, which is then shuttled into the nucleus and binds to p53. βS also suppresses the early-stage aggregation of αS (**C**) by blocking the monomer to dimer transition and secondary nucleation of αS. βS sequesters copper and mitigates the formation of toxic copper-induced αS oligomers (**D**). Aggregated αS inhibits the 26S proteasome, but prior incubation with βS overrides this inhibition (**E**). αS oligomers are cleared by the autophagy-lysosomal pathway. P123H and V70M βS mutations have been linked to lysosomal dysfunction (**F**). Outside the neuron, T cells reactive against L93-L111 of βS are recognized by the MHC class II complex of antigen-presenting cells such as microglia in the brain. The activated T-cells promote neuroinflammation, resulting in neurodegeneration (**G**). This diagram was created with BioRender.com (license agreement number PT239C7U6L, access date 30 November 2021).

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
