# Peer review of "β-Synuclein: An Enigmatic Protein with Diverse Functionality"

_biomolecules, 2022, doi:10.3390/biom12010142_

Round 1
Reviewer 1 Report
Journal: Biomolecules
Manuscript ID: biomolecules-1509726
Title: β-Synuclein: an enigmatic protein with diverse functionality
Authors: Junna Hayashi , John A Carver
This review paper summarizes the literature and authors' data on the structure and functional significance of β-synuclein, which belongs to the synuclein family, which additionally includes α-synuclein and γ-synuclein. The authors provide extensive data, obtained mainly on cells, on a wide range of intracellular β-synuclein-dependent processes in norm and pathology. Some specific aspects of β-synuclein action - association with α-synuclein and dose-dependent qualitative changes, are emphasized. Thus, β-synuclein in a small dose has a protective effect, and in a large dose it exerts a cytotoxic effect. This is the first review to focus specifically on β-synuclein. Indeed, numerous previous reviews have focused primarily on α-synuclein.
Major points.
Despite the overall rather positive impression of the review, it is desirable to supplement it. Thus, despite a wide range of cells and physiological processes, which are regulated by synucleins, including β-synuclein, these proteins play a particularly important role in the regulation of the functioning of nigrostriatal dopaminergic neurons, which are a key link in the regulation of motor function. However, synucleins, including β-synuclein, under certain conditions, become toxic to nigrostriatal dopaminergic proteins, which is one of the most important molecular mechanisms of the pathogenesis of Parkinson's disease. Interesting data are accumulated in the literature on the role of synucleins, including β-synuclein, in the normal regulation of dopamine neurotransmission - vesicular dopamine storing, specific dopamine reuptake, etc. Of great interest are also literature data on the role of β-synuclein in neurodegeneration of nigrostriatal and dopaminergic neurons, e.g., on the structural and functional relationships with specific toxins of dopaminergic neurons (MPTP-MPP +). The development of transgenic mice with a gene knockout for all synucleins made a great contribution to study the role of β-synuclein in neuronal functioning. It is also desirable to briefly mention the relationship between β-synuclein and γ-synuclein, although such data are scarce.
Thus, the paper can be reconsidered after a major revision.
Author Response
We thank the reviewer for their in-depth feedback. The reviewer has made a very relevant suggestion to include a discussion of the role of beta-synuclein in the nigrostriatal system which is selectively affected in Parkinson’s disease. Accordingly, we have included a new paragraph entitled ‘Beta-synuclein regulates the nigrostriatal dopaminergic system’ within the section ‘Beta-synuclein regulates synaptic function, lipid binding and dopamine neurotransmission’ (lines 312-350). We have also updated the Abstract to include ‘the nigrostriatal dopaminergic system’ as a role of beta-synuclein (lines 15-16).
Regarding the relationship between beta-synuclein and gamma-synuclein, as the reviewer states, there is little literature around about gamma-synuclein. However, in the revised text, we have discussed the effects of membrane association of the three synucleins on each other (lines 233-235). Furthermore, we have briefly mentioned the functional compensation between the synucleins in lines 59-63. These conclusions arose from synuclein knockout experiments. Also, lines 75-77 now contain a brief discussion about the compensatory nature of each of the three synucleins.
Overall, we are grateful for the feedback from the reviewer as inclusion of their suggestions has substantially improved our review on the structure, function and interactions of beta-synuclein.
Reviewer 2 Report
The manuscript entitled “β-Synuclein: an enigmatic protein with diverse functionality “presents a very extensive and interesting review about β-synuclein.
The structure of the manuscript is very adequate and the references extensive and updated.
A minor detail. Page 7. Line 313. Authors talk about figure 3. From my point of view this figure is missing. It does not correspond to figure 3 of the manuscript. Probably a new figure about the binding sites of β-synuclein and metal should be included and rename figure 3 to figure 4. It will be easier to follow the text when in line 315 authors talk about M1 and later oxygens of D2.
Author Response
We thank the reviewer for their positive feedback and for picking up on the minor point in relation to Figure 3. We agree that reference to metal binding sites in beta-synuclein is not appropriate for this figure. Instead, in Figure 1, which illustrates the amino acid sequence alignment for alpha-, beta- and gamma-synucleins, we have placed inverted triangles at the position of residues that bind to metals. We have confirmed that all figures are referred to at their appropriate positions within the text.
Round 2
Reviewer 1 Report
ч
The paper has been improved in accordance with most of the reviewer’s recommendations and can be accepted.